# Cost-effectiveness analysis of domiciliary topical sevoflurane for painful leg ulcers

Carmen Selva-Sevilla[1], F. Dámaso Fernández-Ginés[2,¤a], Manuel Cortiñas-Sáenz[3,¤b], Manuel Gerónimo-Pardo[4]*

**1** Department of Applied Economics, Faculty of Economics, University of Castilla La Mancha, Albacete, Spain, **2** Department of Hospital Pharmacy, Torrecárdenas Hospital Complex, Almería, Spain, **3** Unit of Pain —Department of Anesthesiology, Torrecárdenas Hospital Complex, Almería, Spain, **4** Department of Anesthesiology, Complejo Hospitalario Universitario, Albacete, Spain

¤a Current address: Department of Hospital Pharmacy, Hospital La Inmaculada, Huércal Overa, Almería, Spain
¤b Current address: Department of Anesthesiology, Hospital Universitario Virgen de las Nieves, Granada, Spain
* sergepu@hotmail.com, mgeronimop@sescam.jccm.es

**Data Availability Statement:** All relevant data are within the paper and its Supporting Information files.

## Abstract

### Objectives

The general anesthetic sevoflurane is being repurposed as a topical analgesic for painful chronic wounds. We conducted a Bayesian cost-effectiveness analysis (CEA) comparing the addition of domiciliary topical sevoflurane to conventional analgesics (SEVOFLURANE, n = 38) versus conventional analgesics alone (CONVENTIONAL, n = 26) for the treatment of nonrevascularizable painful leg ulcers in an outpatient Pain Clinic of a Spanish tertiary hospital.

### Methods

We used real-world data collected from charts to conduct this CEA from a public healthcare perspective and with a one-year time horizon. Costs of analgesics, visits and admissions were considered, expressed in €2016. Analgesic effectiveness was measured with SPID (Sum of Pain Intensity Difference). A Bayesian regression model was constructed, including "treatment" and baseline characteristics for patients ("arterial hypertension") and ulcers ("duration", "number", "depth", "pain") as covariates. The findings were summarized as a cost-effectiveness plane and a cost-effectiveness acceptability curve. One-way sensitivity analyses, a re-analysis excluding those patients who died or suffered from leg amputation, and an extreme scenario analysis were conducted to reduce uncertainty.

### Results

Compared to CONVENTIONAL, SEVOFLURANE was associated with a 46% reduction in costs, and the mean incremental effectiveness (28.15±3.70 effectiveness units) was favorable to SEVOFLURANE. The estimated probability for SEVOFLURANE being dominant was 99%. The regression model showed that costs were barely influenced by any covariate, whereas effectiveness was noticeably influenced by "treatment". All sensitivity analyses

**Funding:** The authors received no specific funding for this work.

**Competing interests:** Dr. Selva-Sevilla and Dr. Cortiñas-Sáenz have nothing to disclose. Dr. Fernández-Ginés has a patent "Analgetic microspheres comprising a volatile halogenated anaesthetic" issued to Dámaso Fernández-Ginés. Dr. Gerónimo-Pardo owned stock options from Vapogenix Inc, and he had a personal interest in the issue since he started the repurposing process for the inhalation general anesthetic sevoflurane to be used as a topical analgesic. This does not alter our adherence to PLOS ONE policies on sharing data and materials.

showed the robustness of the model, even in the extreme scenario analysis against SEVOFLURANE.

## Conclusions

SEVOFLURANE was dominant over CONVENTIONAL as it was less expensive and much more effective.

## Introduction

Chronic wounds are a major health problem. In a recent systematic review of the literature, chronic wounds of mixed etiologies showed a pooled prevalence of 2.21 per 1000 population; among them, chronic leg ulcers were by far the most frequent type with an estimated prevalence of 1.51 per 1000 population [1].

The available information strongly suggests that chronic wounds represent a great economic burden [1–4]. In a recent systematic review of the literature, a global mean cost of US$ 23300 per patient and year was found (adjusted to 2015 US$), considering a health-care public payer perspective; concerning leg ulcers, and under these same assumptions, the mean cost per patient was US$ 11503 [2].

Among the symptoms of chronic wounds, pain was judged by patients as "the worst symptom and the cause of enormous suffering" [5]. Thus, the treatment of painful chronic leg ulcers should be oriented not only towards healing but also towards alleviating pain [6]. Analgesic palliative treatment becomes particularly relevant for ulcers not expected to heal, such as non-revascularizable vascular leg ulcers [6]. Conventional analgesic drugs remain the mainstay for analgesic treatment of painful chronic wounds, mainly applying a combination of nonsteroidal anti-inflammatory drugs and adjuvants, but some refractory patients need to be treated with the specialized techniques provided by Pain Clinics. Even patients treated in these specialized settings are at risk of developing undesired effects related to analgesic drugs or complications secondary to invasive techniques [6–8].

The best way to prevent these adverse events and complications—and to reduce the associated economic burden—would be to treat painful chronic wounds with useful topical analgesics [8]. Sevoflurane is a well-known ether-derivative inhalational general anesthetic. In recent years, an emerging body of evidence has reported on the analgesic properties of sevoflurane when it is irrigated on the bed of painful leg ulcers [9–18] and wounds of several etiologies [19–21], making it an interesting new alternative for the treatment of leg ulcers.

Concerning economic evaluations in the field of chronic wounds, healing has frequently been selected as the main indicator of effectiveness, followed by the wound-free period and assessments of quality of life [22, 23], but pain has been barely considered. In the context of a randomized clinical trial, a cost-effectiveness analysis (CEA) comparing usual care for chronic leg ulcers versus a new model of community nursing care was conducted; in addition to the number of ulcers healed, the reduction in pain score was also chosen as a primary health outcome [24]. Except for this study, and as far as we know, CEAs focused on the assessment of how pain reduction could reduce the economic burden derived from painful chronic wounds are lacking [22].

As stated before, sevoflurane is showing a great clinical effectiveness when employed as a topical analgesic. However, economic evaluations of this new analgesic alternative have not been conducted so far. The main goal of this study was to conduct a CEA comparing

conventional analgesics alone to treatment including domiciliary topical sevoflurane in addition to conventional analgesics for patients suffering from nonrevascularizable chronic painful leg ulcers who were referred to a Pain Clinic. This study could help healthcare decision-makers to better place sevoflurane as a topical analgesic, based on both clinical and economic criteria.

## Methods

### Background

Patients suffering from refractory pain caused by nonrevascularizable vascular chronic leg ulcers were referred to a Pain Clinic for specialized pain management. Usual analgesic treatment was mainly based on conventional systemic analgesic drugs (non-steroidal anti-inflammatory drugs, opioids, and adjuvants). In addition to conventional analgesics, patients not presenting any safety concern (mainly not having small children living in the same house, and not suffering from any mental disorders), were offered to be treated with *off-label* domiciliary topical sevoflurane following a specific protocol. Details about this protocol are provided in the S1 File.

### Study design, setting, ethical issues, and sample size

This retrospective observational cohort study was conducted in a specialized Pain Clinic of a Spanish tertiary hospital after obtaining approval from the Institutional Review Board (Internal Code 56/2017) and in accordance with the Declaration of Helsinki. Due to the retrospective nature of the research, we were exempted by the same Institutional Review Board of the Complejo Hospitalario Torrecárdenas from asking the patients for informed consent.

A sample size was not calculated due to the retrospective nature of the study. Therefore, the power of the study to detect as statistically significant the difference found between groups in the primary outcome variable was addressed in a *post hoc* analysis using GRANMO v7.11 program (Institut Municipal d'Investigació Mèdica, Barcelona, Spain).

### Target population

Adult patients referred to the Pain Clinic due to painful ulcers from January 2013 to December 2016 were eligible for this study. Patients suffering from nonvascular ulcers and patients not treated on an outpatient basis were excluded. All other patients were included for analysis.

### Comparators

Study groups were made up of patients who received usual analgesic treatment (CONVENTIONAL) and patients who received topical sevoflurane in addition to usual care (SEVOFLURANE).

### Time horizon

For CEAs focused on the healing of chronic wounds, long-term timeframes are usually chosen, frequently a one-year time horizon [22, 23], to account as much as possible for circumstances that could arise during the slow process of healing [22]. Thus, a one-year time horizon was also chosen for the present research, although it was focused on pain rather than healing.

### Data source

Clinical charts were reviewed to collect information on the patients' demographics and the characteristics of their ulcers during the year of follow-up, including pain and analgesics

consumption. Additionally, consumption of any other health resources, such as visits for consults or hospital admissions for any reason, were also collected.

## Study perspective

The perspective was that of the payer (public healthcare system); therefore, only direct medical costs were considered.

## Choice and measurement of health outcomes

Pain reduction was chosen as the primary health outcome for effectiveness. Sum of Pain Intensity Difference (SPID) was the indicator selected to summarize the evolution of pain scores from the first to the last visit during the year of follow-up. SPID was obtained as follows: pain scored at every visit (rated according to the Numerical Rating Scale [NRS]) was subtracted from the pain scored at the first visit to the Pain Clinic, and the result was multiplied by the fraction of time elapsed since every previous evaluation [25]. Hence, the higher the SPID, the greater the effectiveness.

Secondary indicators of health outcomes were leg amputation rate; patients' mortality rate; ulcer healing rate; area of main ulcer at the 12th month; absolute NRS-points of pain at the 12th month; pain-reduction rate; and opioid consumption due to pain, expressed as oral morphine equivalents [26].

## Health resources consumed and related costs

The annual cost per patient was calculated by multiplying the health resources consumed (measured in natural units) by its specific unit costs. All costs were adjusted to € 2016 by using the Consumer Price Index [27]. Since the time horizon was one-year, no discount rate was applied.

For each patient, health resources collected were consumption of sevoflurane and other analgesics, and visits for consults and hospital admissions. All types of visits and hospitalizations were considered for the base-case analysis regardless of whether they could be directly related to the ulcers or not, to be sure not to miss costs derived from unexpected drug adverse effects or from any other unexpected complications related to the ulcers.

To calculate unit costs, three main health resources were taken into account: 1) sevoflurane consumption; 2) consumption of conventional analgesics and adjuvants; and 3) consumption of other health resources as outpatient consultations and hospital admissions.

1. To calculate unit costs related to sevoflurane consumption, three items were considered. First, the cost of a 250mL-bottle of sevoflurane, which was provided by the Pharmacy Department. Second, the cost of the 10 mL-syringes used, which was provided by the Administrative Department of the hospital. And third, the costs derived from the working time spent by a pharmacist to fill a 10 mL-syringe with sevoflurane; this item was estimated from health workers' salaries [28–31] under the assumption that the pharmacist will spend one hour to charge 200 syringes, and the cost for one working hour would be of 2016 €41.

2. The unit costs for conventional analgesics and adjuvants were calculated from the Database of the Knowledge Health Bot PLUS [32].

3. The unit costs for outpatient consults and hospital admissions were obtained from the officially published public health prices [33].

The unit hospitalization costs related to the base case used were based on Spanish Medicare diagnosis-related groups (DRGs). For some diagnoses, DRGs consider two different costs

depending on whether the patients suffered or not from complications while admitted; for such diagnoses, the specific cost with or without complications was attributed to each patient in a case-to-case basis after reviewing patients' charts.

## Assumptions

For patients lacking a real pain score at the end of the follow-up period because they did not complete the period of follow-up because of death or leg amputation, the last available pain score was taken as the final pain score.

Costs derived from cleansing the ulcers at the Pain Clinic were not included for analysis because they were supposed to be a common expenditure for both groups and, besides, they were considered to be included in the costs derived from the visit itself.

## Cost-effectiveness analysis (CEA)

This CEA was not carried out in a frequentist approach but in a Bayesian framework, which allows for the incorporation of a priori information, if available, and to interpret the results in terms of probability [34–38]. To do this, guidelines published by the International Society for Pharmacoeconomics and Outcomes Research to improve the quality of health economic evaluations (Consolidated Health Economic Evaluation Reporting Standards; see S1 Table) [39, 40], retrospective studies [41] and real-world data studies [42] were followed. In addition, the STROBE guidelines were also followed [43].

Taking into consideration that the work was based on observational data, a selection bias could not be discarded. Thus, we planned to perform a linear regression analysis to ascertain if the main results could have been influenced by the assignation to the treatment groups. Specifically, a multiple regression analysis was conducted to identify the part of the difference in costs or effectiveness between groups that was not attributable to the treatment but instead to other covariates, specifically, baseline characteristics of the patients [44–51]. Therefore, covariates allow to reduce the bias and uncertainty of the estimation of the parameters, even if the treatment groups have similar characteristics [48–51]. The specification of this regression model was properly evaluated as it fitted the checklist developed to assess statistical methods for addressing selection bias in CEAs based on observational data [52].

The final model included the following six covariates: *AHT* (arterial hypertension, 0 = No, 1 = Yes); *UlcerDuration*, expressed as months; *UlcerNumber*, measured from 1 to 4 ulcers as this variable was truncated at 4 ulcers; *UlcerDepth*, dichotomized into superficial (0) and profound ulcers (1), being the epidermis and dermis the limit for superficial; *UlcerPain*, expressed as points of pain scored on the NRS, which ranges from 0 (no pain at all) to 10 (the worst imaginable pain); and *Treatment* (SEVOFLURANE = 1, CONVENTIONAL = 0). Details about variable selection are provided in the S2 File.

Thus, for the patient *i*, the linear regression model to explain *Costs* and *Effectiveness*, measured by *SPID*, were as follows:

$$Costs_i = \beta_{11} + \beta_{12} \times AHT_i + \beta_{13} \times UlcerDuration_i + \beta_{14} \times UlcerNumber_i + \beta_{15} \times UlcerDepth_i + \beta_{16} \times UlcerPain_i + \beta_{17} \times Treatment_i + \varepsilon_{1i} \qquad (1)$$

$$Effectiveness_i = \beta_{21} + \beta_{22} \times AHT_i + \beta_{23} \times UlcerDuration_i + \beta_{24} \times UlcerNumber_i + \beta_{25} \times UlcerDepth_i + \beta_{26} \times UlcerPain_i + \beta_{27} \times Treatment_i + \varepsilon_{2i} \qquad (2)$$

To conduct this Bayesian CEA, a multivariate normal distribution for effectiveness and log-transformed total costs was assumed; costs were log-transformed as they were asymmetrically distributed. Additionally, non-informative proper priors were considered, in particular a

multivariate normal prior distribution with a mean vector of zeros and a covariance matrix $10^5$ x Identity-matrix(7) for the vector of means, and a Wishart distribution with 2 degrees of freedom and covariance matrix Identity-matrix(2) was considered as the prior distribution for the variance-covariance matrix.

The expected means for the *Costs* and *Effectiveness* and for all of the coefficients of the model were estimated from the posterior distributions, as well as their 95% Bayesian credible intervals. The estimation of the posterior distribution of the coefficients was performed using Markov Chain Monte Carlo (MCMC).[53, 54] A first burn-in sample with 10,000 simulations was calculated and then discarded. Further, 100,000 simulations were run from which the main statistics of the coefficients were calculated. CEA was conducted using OpenBUGS. The S3 File contains the OpenBUGS codes for our model.

Usually, β–coefficients for the covariate *Treatment* can be interpreted as the incremental cost ($\beta_{17}$) in Eq (1) and the incremental effectiveness ($\beta_{27}$) in Eq (2). However, this interpretation is invalid in the case where a variable was log-transformed, as it was the case for our variable *Costs*. Therefore, incremental cost cannot be considered as the value of the β–coefficient for *Treatment* ($\beta_{17}$). Under these conditions, the costs ratio (costs of SEVOFLURANE divided by costs of CONVENTIONAL) is preferred over the incremental cost (costs of SEVOFLURANE minus costs of CONVENTIONAL). The costs ratio can be obtained from the exponential transformation of the β–coefficient associated with the covariate *Treatment*, namely, exp ($\beta_{17}$). Thus, relative incremental cost attributed to sevoflurane treatment, the treatment under evaluation compared to usual care, can be expressed as (exp ($\beta_{17}$) − 1) x 100.

If needed, the incremental cost-effectiveness ratio (ICER) would be calculated as the ratio between the incremental cost and the incremental effectiveness, expressed as euros per natural unit of effectiveness gained.

Finally, an x-y scatterplot was built to graphically show the posterior costs ratio and incremental effectiveness (the cost-effectiveness plane). Moreover, the probability of a preference for sevoflurane treatment was displayed as a function of the willingness to pay for increasing effectiveness in a natural unit (the cost-effectiveness acceptability curve).

## Sensitivity analyses

To determine the robustness of the model, three types of sensitivity analyses were conducted:

1. To conduct *one-way sensitivity analyses*, the costs of hospitalization was the variable selected because it had the potential for a high uncertainty and, besides, costs derived from hospitalization represented the highest percentage of the total costs. Therefore, base-case cost-effectiveness analysis (CEA) was repeated after calculating the costs derived from hospitalization in three different ways:

   a. The hospitalization costs were based on the Spanish Medicare diagnosis-related groups (DRGs), as for the base-case CEA, yet the costs for each admission were calculated in two ways: first, considering that all patients had suffered from complications while admitted; second, considering that no patient suffered from such complications.

   b. The hospitalization costs were based on the Spanish Medicare DRGs and each admission was assigned a cost with or without complications on a case-to-case basis, as for the base-case CEA, yet only admissions attributed to two specific conditions were considered: admissions attributed to the ulcers by any reason, or admissions attributed to the pain caused by the ulcers.

c. Instead of costs based on the Spanish Medicare DRGs, costs for each admission were calculated by multiplying the cost assigned to every day of hospitalization [33] by the length of stay.

2. Second, the base-case CEA was repeated after excluding the *subgroup of patients* who experienced a major negative outcome before completing a year of follow-up, namely, eight patients who died and three patients whose legs were amputated. Pain was the main clinical variable and, for those patients, pain from the event until the year of follow-up was not measured but extrapolated.

3. Lastly, *extreme scenario analyses* were conducted considering the worst scenario for SEVOFLURANE: values for the variable *Costs* were increased by 25% for SEVOFLURANE and reduced by 25% for CONVENTIONAL, whereas values for the variable *Effectiveness (*SPID*)* were reduced by 25% for SEVOFLURANE and increased by 25% for CONVENTIONAL. The concrete value of 25% was selected *post hoc* after rounding by an excess of the limits of the 95% confidence intervals found for *Costs* and *Effectiveness* (SPID*)*.

## Statistical analyses

Categorical variables are summarized as number and percentage; chi-square tests or Fisher's exact tests were used to explore associations between variables.

Continuous variables are expressed as mean (SD). Student's t-tests or Mann-Whitney U tests were used for comparisons according to the results of the Shapiro-Wilks normality test.

The level of significance was pre-established at 0.05. All of these exploratory statistical analyses were conducted using SPSS Statistics Version 24 (IBM SPSS Statistics, Version 22.0. Armonk, NY, USA). All relevant data are presented in S1 Dataset.

# Results

Seventy-five patients were referred to the Pain Clinic due to painful wounds during the study period. Nine of them were excluded because they suffered from painful wounds that were not to leg ulcers. Two patients suffering from painful leg ulcers were excluded because they were not treated on an outpatient basis.

Overall, 64 patients were included for analysis. The SEVOFLURANE group was composed of 38 patients treated with domiciliary topical sevoflurane in addition to usual care, while the CONVENTIONAL group was composed of 26 patients treated only with usual care.

## Demographic characteristics

Baseline characteristics of patients, leg ulcers, and pain were very similar between groups (Table 1).

## Health outcomes

SEVOFLURANE patients showed a nonsignificant higher percentage of ulcer healing and a significant reduction in ulcer size.

Concerning pain, the SEVOFLURANE group presented significantly better results for all indicators of clinical effectiveness (Table 2).

Accepting an alpha risk of 0.05 in a two sided test and using the widest standard deviation found in the SEVOFLURANE group as the common standard deviation for both groups (18.1), the power of the study was 100% to recognize as statistically significant the striking difference in SPID between groups.

**Table 1. Baseline characteristics.**

|  | SEVOFLURANE (n = 38) | CONVENTIONAL (n = 26) | P-value |
|---|---|---|---|
| **PATIENTS** |  |  |  |
| Age (y) | 70.4 (12.2) | 71.9 (11.2) | 0.681[a] |
| Sex, Female (%) | 63 | 54 | 0.456[b] |
| Weight (kg) | 69.5 (13.2) | 71.85 (18.3) | 0.869[a] |
| Diabetes mellitus (%) | 63 | 81 | 0.130[b] |
| Arterial hypertension (%) | 71 | 77 | 0.602[b] |
| **LEG ULCERS** |  |  |  |
| Etiology, ischemic component (%) | 92 | 96 | 0.640[c] |
| Duration (months) | 24.8 (32.2) | 16.2 (14.5) | 0.864[a] |
| Ulcers per patient (n) | 2.4 (2.3) | 2.8 (2.6) | 0.364[a] |
| Depth beyond dermis (%) | 26 | 31 | 0.697[b] |
| Area, main ulcer (cm$^2$) | 9.2 (11.3) | 10.1 (9.6) | 0.430[a] |
| **ULCER-RELATED PAIN** |  |  |  |
| Pain (NRS) | 6.7 (1.6) | 6.9 (1.4) | 0.797[a] |
| Neuropathic pain (%) | 53 | 62 | 0.481[b] |
| Daily analgesics, active principles (n) | 3.0 (1.2) | 3.0 (1.0) | 0.732[a] |
| Patients taking any opioid (%) | 87 | 85 | 1.000[c] |
| OME only for patients taking opioids (mg/d) | 118.1 (74.6) | 123.1 (89.1) | 0.836[a] |
|  | (n = 33) | (n = 22) |  |

Quantitative data are expressed as the mean (SD). NRS, Numerical Rating Scale; OME, Oral Morphine Equivalent.

[a]Mann-Whitney U test.

[b]Chi square test.

[c]Fisher exact test.

## Costs analysis

Costs of analgesic drugs were significantly higher for the SEVOFLURANE group due to sevoflurane consumption, but the total costs of health resources consumed were significantly lower for SEVOFLURANE because patients in this group consumed significantly fewer health resources (Table 3).

## Cost-effectiveness analysis

The Bayesian cost-effectiveness analysis for the base-case showed lower costs and higher effectiveness (SPID) for the SEVOFLURANE group (Table 4); consequently, this group was dominant over the CONVENTIONAL group.

Regarding costs, *ceteris paribus*, SEVOFLURANE was associated with a reduction of 46% in costs compared to CONVENTIONAL, as indicated by the mean value of the costs ratio (0.54 ±0.15) with a posterior 95% Bayesian credible interval of 0.33 to 0.83 (Table 4).

Regarding effectiveness, *ceteris paribus*, patients in the SEVOFLURANE group experienced an increase of 28.15±3.70 natural units of effectiveness over the CONVENTIONAL, as indicated by the value of the $\beta_{27}$ coefficient, which usually allows for estimating the incremental effectiveness; moreover, this increment was relevant since its posterior 95% Bayesian credible interval was 19.52 to 31.31 (Table 4).

Concerning costs in the regression model, the covariates *UlcerDuration*, *UlcerNumber*, *UlcerDepth*, and *Treatment* showed relevant explanatory power over *Costs* (Table 5). Specifically, the longer the duration of the ulcer, the greater the number of ulcers, and the deeper the

**Table 2. Health outcomes for the follow-up period (12 months).**

|  | SEVOFLURANE (n = 38) | CONVENTIONAL (n = 26) | P-value |
|---|---|---|---|
| **PATIENTS** |  |  |  |
| Patients admitted for any reason (%) | 50 | 85 | 0.007[b] |
| Patients whose admissions were attributed to the ulcers (%) | 21 | 54 | 0.009[b] |
| Patients whose admissions were attributed to ulcer pain (%) | 8 | 38 | 0.004[b] |
| Leg amputation rate (%) | 5 | 4 | 1.000[c] |
| Mortality rate (%) | 11 | 15 | 0.705[c] |
| **LEG ULCERS** |  |  |  |
| Healing rate (%) | 40 | 27 | 0.299[b] |
| Area, main ulcer (cm$^2$) | 5.3 (12.2) | 8.1 (10.2) | 0.018[a] |
| **ULCER-RELATED PAIN** |  |  |  |
| Pain (NRS) | 1.1 (1.0) | 4.1 (2.4) | <0.001[a] |
| Pain reduction rate (%) | 84.1 (14.3) | 39.1 (35.5) | <0.001[a] |
| Patients taking any opioids (%) | 24 | 50 | 0.036[b] |
| OME only for patients taking opioids (mg/d) | 51.9 (44.8) | 144.5 (65.7) | 0.001[a] |
|  | (n = 9) | (n = 13) |  |
| SPID | 41.5 (18.1) | 13.1 (10.4) | <0.001[a] |

Quantitative data are expressed as the mean (SD). NRS, Numerical Rating Scale; OME, Oral Morphine Equivalent; SPID, Sum of Pain Intensity Difference.

[a]Mann-Whitney U test.

[b]Chi square test.

[c]Fisher exact test.

ulcers, the higher the costs. In contrast, total costs were lower for patients treated with topical sevoflurane.

Concerning effectiveness in the regression model, the covariates showing relevant explanatory power over *Effectiveness* were *UlcerDuration*, *UlcerPain* and *Treatment* (Table 5). Specifically, the longer the duration of the ulcer and the higher the baseline pain, the higher the SPID, which implies more pain reduction; but the most remarkable pain reduction was found for patients who had been treated with sevoflurane ($ß_{27}$ = 28.15).

The Bayesian probability for SEVOFLURANE being less expensive was found to be as high as 99%, whereas the probability for SEVOFLURANE being more effective in reducing pain was 100%. Overall, the estimated probability for SEVOFLURANE being dominant was noticeably high (99%). Graphically, the preference for sevoflurane treatment over usual care is shown in the cost-effectiveness plane (Fig 1) and in the cost-effectiveness acceptability curve (Fig 2).

Scatterplot showing the posterior costs ratio and incremental effectiveness measured by the natural unit of effectiveness. SPID, Sum of Pain Intensity Difference.

The curve shows the cost-effectiveness probabilities for SEVOFLURANE by different degrees of willingness to pay for the natural unit of effectiveness. SPID, Sum of Pain Intensity Difference.

## Sensitivity analysis

First, SEVOFLURANE was found to be the dominant alternative in all one-way sensitivity analyses conducted; noteworthy, the probability for SEVOFLURANE being dominant was always higher than 80%.

After excluding eight patients who died and three patients who suffered from leg amputation during the year of follow-up, the final subgroups for the second type of sensitivity analyses

**Table 3. Health resources with their unit costs and associated total health costs.**

| | HEALTH RESOURCES (n) | | | UNIT COSTS (2016 €) | COSTS (€) | | |
|---|---|---|---|---|---|---|---|
| | SEVOFLURANE (n = 38) | CONVENTIONAL (n = 26) | P-value | | SEVOFLURANE (n = 38) | CONVENTIONAL (n = 26) | P-value |
| **Sevoflurane (mL)** | 8784.0 (14116.5) | — | — | 0.30 | 3240 (5657) | — | — |
| **10 mL-syringes** | 878.8 (1411.7) | — | — | 0.17 | 150 (242) | — | — |
| **Charging the syringes** | 878.8 (1411.7) | — | — | 0.21[a] | 181 (293) | — | — |
| **COSTS OF SEVOFLURANE TREATMENT** | — | — | — | — | **3572 (6190)** | — | — |
| **Opioids** | * | * | — | * | 382 (421) | 564 (577) | 0.216 |
| **Nonopioids** | * | * | — | * | 139 (155) | 290 (325) | 0.005 |
| **COSTS OF ALL ANALGESICS, INCLUDING SEVOFLURANE** | — | — | — | — | **4093 (6224)** | **854 (680)** | <0.001 |
| **PRIMARY CARE CONSULTATION, SCHEDULED** | | | | | | | |
| **-Medical** | | | | | | | |
| First visit | 1.0 (0.0) | 1.0 (0.0) | 1.000 | 44 | 44 (0.2) | 44 (0.2) | 1.000 |
| Next visits | 2.6 (2.8) | 5.0 (2.3) | <0.001 | 18 | 47 (50) | 89 (41) | <0.001 |
| **-Medical with nurse** | | | | | | | |
| First visit | 0.4 (0.5) | 0.8 (0.4) | <0.001 | 52 | 19 (25) | 41 (21) | <0.001 |
| Next visits | 1.6 (4.3) | 3.1 (3.7) | 0.001 | 23 | 36 (99) | 71 (83) | 0.001 |
| **-Nurse** | 6.1 (5.9) | 10.1 (7.2) | 0.004 | 21 | 129 (124) | 212 (150) | 0.004 |
| **PRIMARY CARE CONSULTATION, NONSCHEDULED** | | | | | | | |
| Simple | 1.2 (1.7) | 2.9 (2.1) | <0.001 | 49 | 58 (83) | 144 (104) | <0.001 |
| Discharge after observation | 0.3 (0.8) | 0.7 (1.3) | 0.078 | 86 | 23 (72) | 60 (109) | 0.078 |
| Referral to the hospital | 0.5 (0.9) | 2.0 (2.9) | 0.003 | 111 | 56 (99) | 218 (328) | 0.003 |
| **HOSPITAL CONSULTATION, SCHEDULED** | | | | | | | |
| First visit | 1.0 (0.0) | 1.0 (0.0) | 1.000 | 114 | 115 (0.6) | 114 (0.5) | 1.000 |
| Next visits | 8.4 (8.9) | 11.7 (11.2) | 0.003 | 55 | 462 (485) | 638 (618) | 0.003 |
| **HOSPITAL CONSULTATION, NONSCHEDULED** | | | | | | | |
| Discharge after consultation | 0.8 (1.5) | 1.5 (1.3) | 0.009 | 144 | 114 (213) | 211 (193) | 0.009 |
| With further admission | 1.1 (1.9) | 1.88 (1.4) | 0.005 | 392 | 435 (759) | 741 (549) | 0.005 |
| **HOSPITAL ADMISSIONS, DRGS** | 1.13 (1.7) | 2.00 (1.3) | 0.004 | Variable[b] | 6900 (10893) | 12692 (9030) | 0.004 |
| **HYPERBARIC CHAMBER** | 2.6 (11.7) | 1.6 (8.3) | 0.744 | 65 | 180 (784) | 113 (578) | 0.794 |
| **COSTS EXCLUDING ANALGESICS** | — | — | — | — | **8618 (12220)** | **15388 (9860)** | **0.001** |
| **TOTAL COSTS** | — | — | — | — | **12701 (15911)** | **16245 (10012)** | **0.038** |

DRGs, Diagnosis-Related Groups. Data are expressed as mean (SD). Mann-Whitney U test for all comparisons.

*See S2 Table for details.

[a] Costs attributed to a pharmacist to charge one single syringe.

[b] Hospitalization costs were based on Spanish Medicare DRGs [33].

were made up of 32 patients for SEVOFLURANE and 21 patients for CONVENTIONAL. After conducting these new analyses, SEVOFLURANE was also found to be the dominant alternative, with an estimated probability of 98%.

Lastly, the lowest probability for SEVOFLURANE to be dominant was found for the worst scenario for sevoflurane, but even so, its probability was as high as 69% (S3 and S4 Tables).

**Table 4. Statistical summary of costs and effectiveness (100,000 simulations MCMC).**

|  | SEVOFLURANE | | CONVENTIONAL | | Incremental difference | |
|---|---|---|---|---|---|---|
|  | Mean (SD) | 95% CrI | Mean (SD) | 95% CrI | Mean (SD) | 95% CrI |
| Costs (€) | 10250 (2194) | (7316; 14230) | 19750 (4915) | (13260; 28750) | 0.54 (0.15) | **(0.33; 0.83)** |
| Effectiveness (SPID) | 41.37 (2.32) | (37.56; 45.16) | 13.22 (2.82) | (8.59; 17.86) | 28.15 (3.70) | **(22.06; 34.22)** |

CrI not including the zero value are highlighted in bold. CrI, Credible Interval; MCMC, Markov Chain Monte Carlo; SPID, Sum of Pain Intensity Difference.

## Discussion

This study was conducted on patients suffering from painful nonrevascularizable chronic leg ulcers. Compared to usual analgesic care alone, the addition of topical sevoflurane was the dominant alternative since it was associated with both a reduction in costs and an improvement in analgesic effect, making it cost-effective. After further analysis, the probability for sevoflurane being the dominant alternative was found to be as high as 99%. Such a remarkable result seemed to not have been influenced by an eventual selection bias when allocating patients, as it will be further discussed in the Limitations section.

Economic evaluations focused on topical sevoflurane were lacking until now, precluding our results for direct comparison. That is logical, considering that this new indication was first reported only 10 years ago [9].

Regarding effectiveness, the probability of SEVOFLURANE being more effective was 100% for the base-case and for all sensitivity analyses, even under the worst scenario for this group. This finding strongly agrees with a growing body of literature on the great clinical effectiveness of topical sevoflurane for painful vascular ulcers and other wounds of different etiologies [9–21]. In a recent retrospective study with a similar design, intensive care patients suffering from painful pressure ulcers and experiencing adverse effects caused by opioids were prescribed topical sevoflurane; compared to opioids alone, the addition of sevoflurane yielded results in

**Table 5. Estimations of the posterior distribution of the β-coefficients for the cost-effectiveness analysis (100,000 simulations MCMC).**

|  |  | Mean (SD) | 95% CrI |
|---|---|---|---|
| **Costs** | $\beta_{11}$ intercept | 7.65 (0.73) | **(6.44; 8.85)** |
|  | $\beta_{12}$ AHT | 0.43 (0.31) | (-0.09; 0.94) |
|  | $\beta_{13}$ UlcerDuration | 0.02 (0.01) | **(0.01; 0.03)** |
|  | $\beta_{14}$ UlcerNumber | 0.30 (0.11) | **(0.12; 0.47)** |
|  | $\beta_{15}$ UlcerDepth | 0.96 (0.32) | **(0.42; 1.49)** |
|  | $\beta_{16}$ UlcerPain | 0.01 (0.09) | (-0.14; 0.17) |
|  | $\beta_{17}$ Treatment | -0.65 (0.28) | **(-1.10; -0.19)** |
|  | Costs ratio (exp $\beta_{17}$) | 0.54 (0.15) | **(0.33; 0.83)** |
| **Effectiveness** | $\beta_{21}$ intercept | -22.44 (9.79) | **(-38.39; -6.32)** |
|  | $\beta_{22}$ AHT | 1.38 (4.19) | (-5.51; 8.25) |
|  | $\beta_{23}$ UlcerDuration | 0.12 (0.07) | **(0.00; 0.23)** |
|  | $\beta_{24}$ UlcerNumber | 0.60 (1.42) | (-1.74; 2.95) |
|  | $\beta_{25}$ UlcerDepth | 1.33 (4.33) | (-5.78; 8.43) |
|  | $\beta_{26}$ UlcerPain | 4.51 (1.26) | **(2.44; 6.58)** |
|  | $\beta_{27}$ Treatment | 28.15 (3.70) | **(22.06; 34.22)** |

CrI not including the zero value are highlighted in bold. AHT, Arterial Hypertension; CrI, Credible Interval; MCMC:,Markov Chain Monte Carlo.

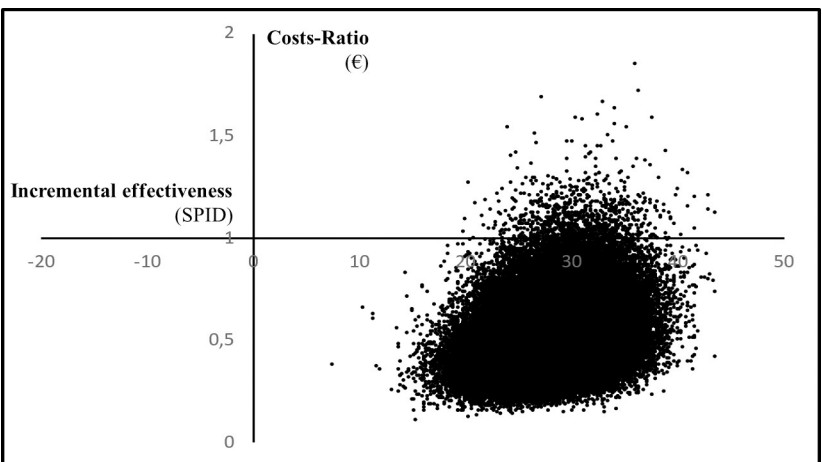

**Fig 1. Cost-effectiveness plane.**

effectiveness similar to ours, as pain scores were significantly lower and opioid consumption was significantly reduced [21].

The addition of sevoflurane increased the costs of analgesic medication despite reducing the consumption of conventional analgesics, as they are usually cheaper than sevoflurane. However, total costs were lower for the SEVOFLURANE group due to remarkably lower consumption of more expensive health resources, namely, visits for consults and hospital admittances. Interestingly, these patients were less frequently admitted for ulcer pain or for any other reason related to their ulcers. Whether the reduction in conventional analgesics consumption was followed by less consumption of health resources due to a further reduction of unwanted events could not be ascertained due to the retrospective nature of the study, but this hypothesis is attractive and deserves further research.

The CONVENTIONAL group was made up of patients referred to the Pain Clinic who did not benefit from sevoflurane due to safety concerns. This problem could be easily solved by treating them at the primary care center [17] or even at home by a domiciliary support team [18]. In our opinion, both strategies would be cost-effective, since the costs arising from the visits would be counterbalanced by the clinical benefits achieved.

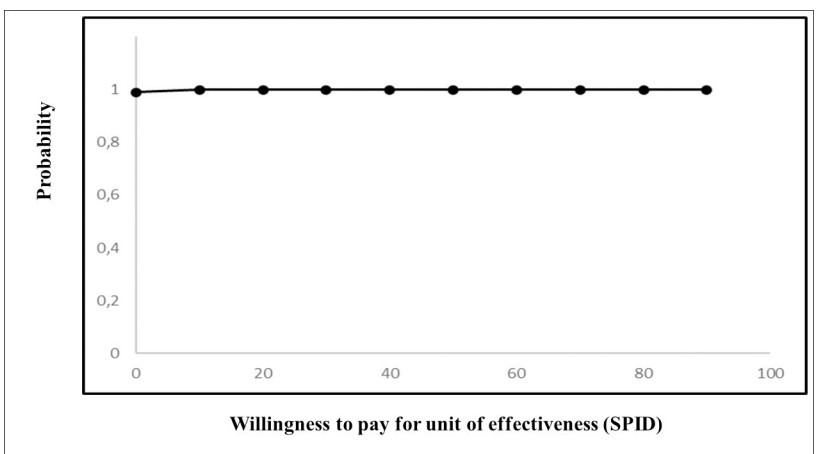

**Fig 2. Cost-effectiveness acceptability curve.**

## Study perspective

As many other authors did [22, 23], we chose a public payer perspective instead of a societal one. Therefore, indirect costs related to the societal perspective, such as those derived from loss of productivity, were excluded from the analysis. In our opinion, a public payer perspective is suitable for several reasons. First, indirect costs may have a little impact on total costs; in a study conducted in Germany describing costs associated with chronic leg ulcers, indirect costs represented only 10% of the total costs [55]. Second, patients suffering from chronic leg ulcers are usually retired because frequently they are elderly, as our patients were; thus, costs derived from loss of productiveness were not applicable [56]. Not surprisingly, the societal perspective has been scarcely applied so far; for instance, the societal perspective was reported in only 14% of CEAs conducted on chronic wounds [22].

As the chosen perspective was that of the public payer, only directs costs were considered for the CEA. We aimed not to miss any cost derived from the consumption of health resources because of the ulcers, and it was a difficult task to properly discriminate which costs were related or unrelated to the ulcers; for this reason, we decided to consider for the base-case analysis all costs derived from all resources collected from the charts. Nevertheless, in order to conduct two sensitivity analyses, costs were reassigned either as related to the ulcers by any reason or related exclusively to the pain caused by the ulcers; the estimated probability for SEVO-FLURANE to be dominant was also high after both re-analyses, specifically 90% and 81%, respectively.

## Strengths

In our opinion, this work has several strengths that deserve comment.

First, CEAs in the field of chronic wounds have been mainly focused on several aspects of healing, while pain has been scarcely studied so far. To the best of our knowledge, this is the first time a CEA conducted in the setting of chronic wounds focused on pain, and our encouraging results strongly suggest that this symptom should be more frequently taken into account in future research.

Second, this was not a clinical trial but an observational study. As such, patients were treated by their attending physician according to the best clinical judgement. Thus, our results are based on real-world data, which allows for generalization of the results to the day-to-day practice [57, 58]. In our opinion, this approach represents an advantage over works that try to model the natural history of the disease based on population data [59].

## Limitations

In addition to the inherent limitations common to all CEAs (see CHEERS checklist in S1 Table in supporting information), the present study has the following particular limitations.

The main limitation of this retrospective observational study is that a selection bias could not be completely ruled out. Safety concerns were the main factor in deciding whether any specific patient was suitable to be treated with sevoflurane. Nonetheless, both groups showed quite similar baseline characteristics, whereas their pain evolution was markedly different from the very beginning. Even so, to try to eliminate any eventual bias and to reduce uncertainty, we conducted a regression model considering those variables that could affect cost and effectiveness to obtain the true effect of the treatment on the CEA result. In addition to *Treatment*, some other variables such as *UlcerDuration*, *UlcerDepth*, and *UlcerPain* showed relevant explanatory power on costs or effectiveness. However, costs were barely influenced by any of these variables, whereas effectiveness was markedly influenced only by *Treatment*, as indicated

by the values of the ß-coefficients (Table 5). Considering this, we do not believe our results were influenced by selection bias, even when accepting that such bias could have existed.

Second, the sample size was relatively small—although quite similar to many other studies on chronic wounds—[22] because of two main reasons. We conducted a Spanish single center study because, currently, there is no other Pain Clinic treating patients with domiciliary topical sevoflurane, precluding the possibility of a multicenter study. The sample size was further limited by the fact that, according to the approved protocol, only patients suffering from nonrevascularizable leg ulcers were suitable to be referred to the Pain Clinic for palliative analgesic treatment. Nevertheless, we found *post hoc* that our study was not underpowered.

Third, this specificity in design could affect the generalization of our findings to other settings or other types of wounds. Generalization to other countries should also be done with caution since unit costs could be different in other socio-economic contexts. However, the present CEA showed robust results favoring SEVOFLURANE, which was mainly due to the remarkable difference in the analgesic effectiveness of topical sevoflurane. Of note, SEVOFLURANE remained dominant even under extreme scenario analysis, which was the worst scenario for this group. Therefore, SEVOFLURANE can be considered dominant to CONVENTIONAL in terms of cost-effectiveness with a high degree of confidence [60].

## Conclusions

In conclusion, the results of this retrospective observational cohort study strongly suggest that the addition of domiciliary topical sevoflurane to the usual care provided by a Pain Clinic for nonrevascularizable painful leg ulcers is a cost-effective alternative compared to usual care alone. The addition of sevoflurane was followed by a reduction in costs and, of note, by a great improvement in analgesic effectiveness. Therefore, healthcare decision-makers should consider this new alternative, as it could lead to a more efficient application of healthcare resources.

## Supporting information

**S1 File. Background on the use of topical sevoflurane.**
(PDF)

**S2 File. Selection of covariates for Bayesian regression model.**
(PDF)

**S3 File. OpenBUGS codes for cost-effectiveness analysis.**
(PDF)

**S1 Table. CHEERS checklist.**
(PDF)

**S2 Table. Mean consumption of pharmaceutical formulations for each group.**
(PDF)

**S3 Table. Sensitivity analyses: Statistical summary of costs and effectiveness (100,000 simulations MCMC).**
(PDF)

**S4 Table. Sensitivity analyses: Estimations of the posterior distribution of the β-coefficients and of the probabilities related to the cost-effectiveness analysis (100,000 simulations MCMC).**
(PDF)

**S1 Dataset.**
(XLSX)

**S1 Striking image.**
(TIF)

## Acknowledgments

The authors thank Fernando Andrés Petrel, M.S. (Clinical Research Support Unit, General University Hospital, Albacete, Spain) for his invaluable help with the statistical analysis; and to Miguel Angel Negrín Hernández, Ph.D. (Department of Quantitative Methods, Universidad de las Palmas de Gran Canaria, Las Palmas, Spain) for his invaluable help with the specification and interpretation of Bayesian regression models for the cost-utility and cost-effectiveness analyses.

## Author Contributions

**Conceptualization:** Carmen Selva-Sevilla, F. Dámaso Fernández-Ginés, Manuel Cortiñas-Sáenz, Manuel Gerónimo-Pardo.

**Data curation:** Carmen Selva-Sevilla.

**Formal analysis:** Carmen Selva-Sevilla, Manuel Gerónimo-Pardo.

**Investigation:** Carmen Selva-Sevilla, F. Dámaso Fernández-Ginés, Manuel Cortiñas-Sáenz, Manuel Gerónimo-Pardo.

**Methodology:** Carmen Selva-Sevilla, F. Dámaso Fernández-Ginés, Manuel Cortiñas-Sáenz, Manuel Gerónimo-Pardo.

**Project administration:** Carmen Selva-Sevilla.

**Resources:** Manuel Cortiñas-Sáenz.

**Software:** Carmen Selva-Sevilla.

**Supervision:** Carmen Selva-Sevilla.

**Visualization:** Carmen Selva-Sevilla, Manuel Gerónimo-Pardo.

**Writing – original draft:** Carmen Selva-Sevilla, Manuel Gerónimo-Pardo.

**Writing – review & editing:** Carmen Selva-Sevilla, F. Dámaso Fernández-Ginés, Manuel Cortiñas-Sáenz, Manuel Gerónimo-Pardo.

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
