## [Decision Letter · Decision Letter 0]

18 Aug 2021

PONE-D-21-09306

Cost-effectiveness analysis of domiciliary topical sevoflurane for painful leg ulcers

PLOS ONE

Dear Dr. Gerónimo Pardo,

Thank you for submitting your manuscript to PLOS ONE. After careful consideration, we feel that it has merit but does not fully meet PLOS ONE’s publication criteria as it currently stands. Therefore, we invite you to submit a revised version of the manuscript that addresses the points raised during the review process.

We look forward to receiving your revised manuscript.

Kind regards,

Carlos Alberto Zúniga-González, Ph.D

Academic Editor

PLOS ONE

Additional Editor Comments:

Dear authors I am suggest you for improvements you manuscript I have checking that the Cost-effectiveness analysis is used to support health sector decisions about the allocation of limited resources with the objective of maximizing health, he subject is very interesting and very pertinent, I would like to suggest a couple of references for you to evaluate them in terms of Cost-effectiveness: 1) Zuniga Gonzalez, Carlos Alberto (2009): Technical efficiency of organic fertilizer in small farms of Nicaragua: 1998-2005. Published in: African Journal of Business Management , Vol. 5, No. 3 (4 February 2011): pp. 967-973. Available in https://publons.com/publon/11272633/ and https://mpra.ub.uni-muenchen.de/49352/ 2) Dios-Palomares, Rafaela; Alcaide, David; Diz, Jose; Jurado, Manuel; Prieto, Angel; Morantes, Martina; Alberto Zuniga, Carlos (2015). Analysis of the Efficiency of Farming Systems in Latin America and the Caribbean Considering Environmental Issues. Revista Cientifica-Facultad de Ciencias Veterinarias, 25(1). Available in https://publons.com/publon/3106827/ 3) Zuniga González, C. (2020). Crecimiento de la productividad total de los factores en la agricultura: análisis del índice de Malmquist de 14 países, 1979-2008. REICE: Revista Electrónica De Investigación En Ciencias Económicas, 8(16), 68-97. https://doi.org/10.5377/reice.v8i16.10661 4) Blanco-Orozco, N., Arce-Díaz, E., & Zúñiga-Gonzáles, C. (2015). Integral assessment (financial, economic, social, environmental and productivity) of using bagasse and fossil fuels in power generation in Nicaragua. Revista Tecnología en Marcha, 28(4), 94-107. Available in https://publons.com/publon/32281799/

Journal Requirements:

[I have read the journal's policy and the authors of this manuscript have the following competing interests:

Dr. Selva-Sevilla and Dr. Cortiñas-Sáenz have nothing to disclose.

Dr. Fernández-Ginés has a patent “Analgetic microspheres comprising a volatile halogenated anaesthetic” issued to Dámaso Fernández-Ginés.

Dr. Gerónimo-Pardo owned stock options from Vapogenix Inc, and he had a personal interest in the issue since he started the repurposing process for the inhalation general anesthetic sevoflurane to be used as a topical analgesic.].

Reviewers' comments:

Reviewer's Responses to Questions

**Comments to the Author**

1. Is the manuscript technically sound, and do the data support the conclusions?

Reviewer #1: Yes

Reviewer #2: Yes

2. Has the statistical analysis been performed appropriately and rigorously? 

Reviewer #1: Yes

Reviewer #2: Yes

3. Have the authors made all data underlying the findings in their manuscript fully available?

Reviewer #1: Yes

Reviewer #2: Yes

4. Is the manuscript presented in an intelligible fashion and written in standard English?

Reviewer #1: Yes

Reviewer #2: Yes

5. Review Comments to the Author

Reviewer #1: This is a small, but well-conducted study and the authors make appropriate use of quantitive techniques as well as presenting their findings in an intelligible way that includes the presentation of the main results in both tabular format and in Figures (e.g., cost-effectiveness planes). I have no real concerns with the methods, but I do think the paper would be improved if the authors were to say more about the possibility of selection bias towards the front of the paper and talk about how they propose to use regression analysis to address this possibility as far as possible, given the retrospective design. At the moment, the discussion of this is relegated to the end of the paper - it would be better to set up the proposed approach to trying to detect selection bias early in the paper as a way of motivating the regression work, in particular.

The manuscript would also benefit from some English-language editing, even though it is generally quite well-written.

Reviewer #2: Chronic wounds represent an unmet medical need and considerable economic burden. Repurposing of sevoflurane as a topical analgesic may provide an alternative to conventional analgesics. Cost-effectiveness analysis is used to support health sector decisions about the allocation of limited resources with the objective of maximising health. In the light of the above, the article appears to be sound, actual and focused.

Some points to consider in subsequent versions:

Minor: The choice of Bayesian statistics includes the statement /lines 197-199/: “This CEA was not carried out in a frequentist approach but in a Bayesian framework, which allows for the incorporation of a priori information and to interpret the results in terms of probability”. But above, line 88 we see: “However, economic evaluations of this new analgesic alternative have not been conducted so far.” May authors clarify where the prior comes from? Is the reference 55 mentioned later /line 450/ constitutes the prior?

Minor: The comparator is “conventional” analgesics. Table 3 has two sections – opioids and non-opioids. The range in opioids may include from tramadol to morphine, non- opioids are much larger in terms of choice. Some patients might have crossed over. Despite the small sample size, it is important to describe more accurately the nature of the conventional analgesics as effects may vary. Line 435 – 437: “Whether the reduction in opioid consumption was followed by less consumption of health resources could not be ascertained due to the retrospective nature of the study”. What about the non- opioids as well as those who might have both?

Minor: It is important to justify the lack of utilities. Indeed, the study is observational and retrospective, however there may well be utilities from the literature, including such collected from PROs in interventional trials. The key advantage is the ability to use a well- defined utility as an outcome measure across illnesses to help decision-makers put health care choices in perspective. I would encourage the authors to elaborate whether SPID has been perceived as utility.

Minor: In the sensitivity analysis we see one- way sensitivity analysis /without Tornado diagram, to explain/. Why probabilistic sensitivity analysis has not been conducted?

Minor: CEA, per se, is associated with well-acknowledged shortcomings, such as stochastic uncertainty, heterogeneity and treatment stratification, representation uncertainty, misinterpretation of censored data, list not exhaustive. Would the authors resonate that in the limitations?

In summary, the paper requires minor revisions.

A confidential comment for the editors: the study is based on off- label use of sevoflurane. What is your policy about potential promotion of off- label use?

6. PLOS authors have the option to publish the peer review history of their article (what does this mean?). If published, this will include your full peer review and any attached files.

Reviewer #1: No

Reviewer #2: **Yes: **Borislav Borissov

---

## [Author Response · Author response to Decision Letter 0]

24 Aug 2021

Additional Editor Comments:

Dear authors I am suggest you for improvements you manuscript I have checking that the Cost-effectiveness analysis is used to support health sector decisions about the allocation of limited resources with the objective of maximizing health, he subject is very interesting and very pertinent, I would like to suggest a couple of references for you to evaluate them in terms of Cost-effectiveness: 

1) Zuniga Gonzalez, Carlos Alberto (2009): Technical efficiency of organic fertilizer in small farms of Nicaragua: 1998-2005. Published in: African Journal of Business Management , Vol. 5, No. 3 (4 February 2011): pp. 967-973. Available in https://publons.com/publon/11272633/ and https://mpra.ub.uni-muenchen.de/49352/

2) Dios-Palomares, Rafaela; Alcaide, David; Diz, Jose; Jurado, Manuel; Prieto, Angel; Morantes, Martina; Alberto Zuniga, Carlos (2015). Analysis of the Efficiency of Farming Systems in Latin America and the Caribbean Considering Environmental Issues. Revista Cientifica-Facultad de Ciencias Veterinarias, 25(1). Available in https://publons.com/publon/3106827/

3) Zuniga González, C. (2020). Crecimiento de la productividad total de los factores en la agricultura: análisis del índice de Malmquist de 14 países, 1979-2008. REICE: Revista Electrónica De Investigación En Ciencias Económicas, 8(16), 68-97. https://doi.org/10.5377/reice.v8i16.10661

4) Blanco-Orozco, N., Arce-Díaz, E., & Zúñiga-Gonzáles, C. (2015). Integral assessment (financial, economic, social, environmental and productivity) of using bagasse and fossil fuels in power generation in Nicaragua. Revista Tecnología en Marcha, 28(4), 94-107. Available in https://publons.com/publon/32281799/

AUTHORS: We thank the Academic Editor for his kind suggestion. We will take those papers in consideration for future studies.

REVIEWER: Reviewer #1: This is a small, but well-conducted study and the authors make appropriate use of quantitive techniques as well as presenting their findings in an intelligible way that includes the presentation of the main results in both tabular format and in Figures (e.g., cost-effectiveness planes). I have no real concerns with the methods, but I do think the paper would be improved if the authors were to say more about the possibility of selection bias towards the front of the paper and talk about how they propose to use regression analysis to address this possibility as far as possible, given the retrospective design. At the moment, the discussion of this is relegated to the end of the paper - it would be better to set up the proposed approach to trying to detect selection bias early in the paper as a way of motivating the regression work, in particular.

AUTHORS: We fully agree with the reviewer in the importance of addressing a selection bias in retrospective studies like ours, based on observational data.

We took it into consideration in order to design a statistical plan to ascertain if our results could have been influenced by such eventual selection bias, as it was stated in the Methods section (lines 204-213), particularly in the sentence supported by reference 52: “The specification of this regression model was properly evaluated as it fitted the checklist developed to assess statistical methods for addressing selection bias in CEAs based on observational data”.

Concerning the Discussion section, we decided that the best part to elaborate on this topic would be the Limitation section, as such decision fits the usual structure of a scientific paper, where Limitations appear at the end, just after Strengths. 

Now, following the reviewer recommendation, we have made some modifications in the text to emphasize the importance of addressing a selection bias.

First, the following sentence has been added at the beginning of the paragraph explaining the methodology for the linear regression analysis (lines 204-207): 

“Taking into consideration that the work was based on observational data, a selection bias could not be discarded. Thus, we planned to perform a linear regression analysis to ascertain if the main results could have been influenced by the assignation to the treatment groups. Specifically, a multiple regression analysis…”.

In our opinion, such specification in the Methods section will help to make the readers more aware of the importance of this topic earlier in the manuscript.

Second, the following sentence has been added to the end of the first paragraph in the Discussion section (lines 420-422): 

“Such a remarkable result seemed to not have been influenced by an eventual selection bias when allocating patients, as it will be further discussed in the Limitations section”. 

As a result, the importance of addressing a selection bias has been emphasized as recommended by the reviewer, so that the readers will be fully aware of it before finding a full explanation in the Limitation section.

We thank the reviewer for the comment.

REVIEWER:The manuscript would also benefit from some English-language editing, even though it is generally quite well-written.

AUTHORS: We are not native English speakers. Thus, the text was revised by American Journal Experts before the first submission to the journal, as stated in the Cover letter. We can provide you with a copy of the certificate under request.

REVIEWER: Reviewer #2: Chronic wounds represent an unmet medical need and considerable economic burden. Repurposing of sevoflurane as a topical analgesic may provide an alternative to conventional analgesics. Cost-effectiveness analysis is used to support health sector decisions about the allocation of limited resources with the objective of maximising health. In the light of the above, the article appears to be sound, actual and focused.

Some points to consider in subsequent versions:

Minor: The choice of Bayesian statistics includes the statement /lines 197-199/: “This CEA was not carried out in a frequentist approach but in a Bayesian framework, which allows for the incorporation of a priori information and to interpret the results in terms of probability”. But above, line 88 we see: “However, economic evaluations of this new analgesic alternative have not been conducted so far.” May authors clarify where the prior comes from? Is the reference 55 mentioned later /line 450/ constitutes the prior?

AUTHORS: The sentence quoted by the reviewer (lines 197-199) describes two general characteristics of the Bayesian methodology, namely the possibility of incorporating a priori information, and also to interpret the results in terms of probability. Regarding priors, a priori information will not always be available for all evaluations, which would be especially true when the research will be focused on a new topic (as it was the case in our paper). In such cases, it seems reasonable to assume a non-informative prior about the parameters of interest and let the sample data to control the posterior distribution. The use of non-informative priors is a usual procedure in the literature, as the following papers exemplify:

- Isetta V, et al; SPANISH SLEEP NETWORK. (2015). A Bayesian cost-effectiveness analysis of a telemedicine-based strategy for the management of sleep apnoea: a multicentre randomised controlled trial. Thorax. 2015 Nov;70(11):1054-61.https://doi.org/10.1136/thoraxjnl-2015-207032. 

- Lugo VM, et al. (2019) Comprehensive management of obstructive sleep apnea by telemedicine: Clinical improvement and cost-effectiveness of a Virtual Sleep Unit. A randomized controlled trial. PLoS ONE 14(10): e0224069. https://doi.org/10.1371/journal.pone.0224069.

- Masa JF, et al; Spanish Sleep Network. (2020). Cost-effectiveness of positive airway pressure modalities in obesity hypoventilation syndrome with severe obstructive sleep apnoea. Thorax. 2020 Jun;75(6):459-467. htpp://doi.org/10.1136/thoraxjnl-2019-213622.

In our opinion, there was not available a priori information before conducting our research. 

Regarding effectiveness, we chose pain as the primary outcome. However, complex chronic wounds have been mostly approached in terms of healing or patients’ quality of life, as stated by the reviews authored by Tricco et al (reference 22) and Carter et al (reference 23). As far as we know, pain was considered as an indicator of effectiveness in an only paper (lines 83-84, reference 24), but it lacked pain reduction data. 

Regarding costs, costs related to health resources could be different between different socioeconomic contexts (line 507) and, thus, they could not be extrapolated to other contexts. Accordingly, the reference cited by the reviewer (reference 55) was not useful as that research was conducted in Germany. Our research was conducted in the context of Spanish health system and, to the best of our knowledge, there are not previous similar research conducted in Spain.

Regarding the new alternative under evaluation, topical sevoflurane, our research represents the first study assessing it analgesic effectiveness for the long-term management of refractory pain caused by chronic ulcers.

Overall, we decided to assume non-informative priors, as stated in line 229.

To improve understanding, we have added a specification in the sentence quoted by the reviewer (lines 197-199): 

“This CEA was not carried out in a frequentist approach but in a Bayesian framework, which allows for the incorporation of a priori information, if available, and to interpret the results in terms of probability [34-38].”

Thank you.

REVIEWER: Minor: The comparator is “conventional” analgesics. Table 3 has two sections – opioids and non-opioids. The range in opioids may include from tramadol to morphine, non- opioids are much larger in terms of choice. Some patients might have crossed over. Despite the small sample size, it is important to describe more accurately the nature of the conventional analgesics as effects may vary.

AUTHORS: Please, find a detailed consumption of opioids and nonopioids drugs for each group in Supplementary Table 2.

Thank you.

REVIEWER: Line 435 – 437: “Whether the reduction in opioid consumption was followed by less consumption of health resources could not be ascertained due to the retrospective nature of the study”. What about the non- opioids as well as those who might have both?

AUTHORS: We focused this part of the discussion on the eventual advantages of reducing opioids consumption as it represents a big health problem, in particular in USA. However, the reviewer’s commentary about non-opioids analgesics is accurate, as limiting their use could also help to reduce health resources related to a decrease in adverse events, such as gastrointestinal hemorrhages or kidney failures. Thus, we have modified the sentence in this sense (lines 440-442):

“Whether the reduction in conventional analgesics opioid consumption was followed by less consumption of health resources due to a further reduction of adverse events could not be ascertained due to the retrospective nature of the study, but this hypothesis is attractive and deserves further research.”

Thank you.

REVIEWER: Minor: It is important to justify the lack of utilities. Indeed, the study is observational and retrospective, however there may well be utilities from the literature, including such collected from PROs in interventional trials. The key advantage is the ability to use a well- defined utility as an outcome measure across illnesses to help decision-makers put health care choices in perspective. I would encourage the authors to elaborate whether SPID has been perceived as utility.

AUTHORS: We fully agree with the reviewer in the importance of performing cost-utility analysis in addition to cost-effectiveness analysis. In fact, we have reported some cost-utility analysis and other papers in which quality of life was assessed (please find them at the end of this response).

However, preference-based health-related quality of life measures like EQ5D or SF36 should be used to generate utilities. The assessment of patients’ quality of life was not a common procedure in the Pain Clinic by the time the patients were treated, precluding us to perform a cost-utility analysis based on our own patients’ data.

We could not use utilities from the literature because, to the best of our knowledge, there are no utilities reported for so specific patients and treatments: patients suffered from painful non-revascularizable vascular leg ulcers, and they received an at-home specialized analgesic treatment which included the employment of topical sevoflurane. 

Instead, we designed a cost-effectiveness analysis in which the evolution of pain over time was considered as the main variable of effectiveness, since patients had been referred to the Pain Clinic by their attending vascular surgeons to receive palliative analgesic treatment and, accordingly, pain was assessed in every visit to the clinic. In this sense, SPID was considered the main variable of outcome. SPID integrates in a single numerical value the evolution of pain over time, one year in our case, but it does not measure patients’ quality of life and, consequently, it cannot be used to calculate utilities. 

In the future, we plan to conduct a prospective research assessing the quality of life of patients treated with conventional analgesic treatment compared to patients also treated with topical sevoflurane. In this future research quality of life will be surveyed through EQ5D and SF36 questionnaires to further calculate their associated utilities and QALYs and, lastly, to perform a cost-utility analysis.

In our opinion, it would not be appropriate to justify the lack of utilities in the text of the article because our main goal was not to conduct a cost-utility analysis but a cost-effectiveness one.

Thank you.

-Selva-Sevilla C, Conde-Montero E, Gerónimo-Pardo M. Bayesian regression model for a cost-utility and cost-effectiveness analysis comparing punch grafting versus usual care for the treatment of chronic wounds. International Journal of Environmental Research and Public Health 2020;17:3823.

-Selva-Sevilla C, Ferrara P, Gerónimo-Pardo M. Cost-utility analysis for recurrent lumbar disc herniation. Conservative treatment versus discectomy versus discectomy with fusion. Clinical Spine Surgery 2019;32:E228–E234.

-Selva-Sevilla C, Ferrara P, Gerónimo-Pardo M. Interchangeability of the EQ‑5D and the SF‑6D, and comparison of their psychometric properties in a spinal postoperative Spanish population. European Journal of Health Economics 2020;21:649-662.

-Cifuentes Tébar J, Rueda-Martínez JL, Selva-Sevilla C, Gerónimo-Pardo M. Analgesic effectiveness and improvement in quality of life after using topical sevoflurane for an extremely painful anal fissure. Journal of Coloproctology 2021;41:206–209.

REVIEWER: Minor: In the sensitivity analysis we see one- way sensitivity analysis /without Tornado diagram, to explain/. 

AUTHORS: In economic evaluations, tornado diagrams are used to present the result of multiple univariate sensitivity analyses on a single graph. Each analysis is summarized using a horizontal bar which represents the variation in the model output (usually an ICER) around a central value (corresponding to the base case analysis) as the relevant parameter is varied between two plausible but extreme values (from https://yhec.co.uk/glossary/tornado-diagram/).

But ICER is exclusively calculated in those cases in which the result of the cost-effectiveness analysis is not definitive, either because it is cheaper but also less effective, or more effective but also more expensive. ICER is not calculated in cases when the alternative under evaluation is either clearly favored (cheaper and more effective) or discarded (more expensive and less effective). 

In our case, topical sevoflurane was the alternative clearly dominant as it was cheaper and much more effective not only in the case base but also in all one-way sensitivity analysis, so there was no need to calculate ICER and to build a tornado diagram. In our opinion, it makes more sense to represent the results of the sensitivity analysis as we did in table S3.

Thank you.

REVIEWER:Why probabilistic sensitivity analysis has not been conducted?

AUTHORS: Traditionally, parameter uncertainty has been examined using sensitivity analysis. There are three main statistic models to assess for uncertainty: bootstrapping, probabilistic sensitivity analysis, and Bayesian analysis. 

We decided to design our research under a probabilistic Bayesian approach. Bayesian perspective permits a natural interpretation of the uncertainty of the results in terms of probability and a robust estimation of the precision of the estimates (reference 50). To do it, Markov Chain Monte Carlo simulation methodology is used, as in probabilistic sensitivity analysis.

We used MCMC (former lines 237-239) and, additionally, we also performed a regression model to handle uncertainty in our CEA (reference 49).

Overall, we did not consider it appropriate to conduct a probabilistic sensitivity analysis.

Thank you for your comment.

REVIEWER: Minor: CEA, per se, is associated with well-acknowledged shortcomings, such as stochastic uncertainty, heterogeneity and treatment stratification, representation uncertainty, misinterpretation of censored data, list not exhaustive. Would the authors resonate that in the limitations?

AUTHORS: As pointed out by the reviewer, cost-effectiveness analyses have general limitations. As a result, the International Society for Pharmacoeconomics and Outcomes Research (ISPOR) published guidelines to acknowledge such limitations and to improve the quality of health economic evaluations. The guideline CHEERS (Consolidated Health Economic Evaluation Reporting Standard) is such an example of this.

As stated in the manuscript (lines 199-203), we followed the guidelines published by ISPOR (references 39 and 40) to conduct our CEA. Additionally, Table S1 in supporting information shows the CHEERS checklist.

After your comment, we have added a paragraph to start the Limitations section (lines 485-486):

“In addition to the inherent limitations common to all CEAs (see CHEERS checklist in Table S1 in supporting information), the present study has the following particular limitations.”

Thank you.

---

## [Decision Letter · Decision Letter 1]

3 Sep 2021

Cost-effectiveness analysis of domiciliary topical sevoflurane for painful leg ulcers

PONE-D-21-09306R1

Dear Dr. Manuel Gerónimo Pardo,

We’re pleased to inform you that your manuscript has been judged scientifically suitable for publication and will be formally accepted for publication once it meets all outstanding technical requirements.

Kind regards,

Carlos Alberto Zúniga-González, Ph.D

Academic Editor

PLOS ONE

Additional Editor Comments (optional):

We appreciate the effort that was made to improve the quality of your manuscript, and I am pleased to accept its publication.

Reviewers' comments:

Reviewer's Responses to Questions

**Comments to the Author**

1. If the authors have adequately addressed your comments raised in a previous round of review and you feel that this manuscript is now acceptable for publication, you may indicate that here to bypass the “Comments to the Author” section, enter your conflict of interest statement in the “Confidential to Editor” section, and submit your "Accept" recommendation.

Reviewer #1: All comments have been addressed

Reviewer #2: All comments have been addressed

2. Is the manuscript technically sound, and do the data support the conclusions?

Reviewer #1: Yes

Reviewer #2: Yes

3. Has the statistical analysis been performed appropriately and rigorously? 

Reviewer #1: Yes

Reviewer #2: Yes

4. Have the authors made all data underlying the findings in their manuscript fully available?

Reviewer #1: Yes

Reviewer #2: Yes

5. Is the manuscript presented in an intelligible fashion and written in standard English?

Reviewer #1: Yes

Reviewer #2: Yes

6. Review Comments to the Author

Reviewer #1: The authors have addressed my comments in a satisfactory way and seem to have answered the other referee's comments too.

Reviewer #2: I have no further comments. Authors modified a number of sentences to adopt the questions asked. Also, limitations and well known shortcomings o the method have been addressed.

7. PLOS authors have the option to publish the peer review history of their article (what does this mean?). If published, this will include your full peer review and any attached files.

Reviewer #1: No

Reviewer #2: **Yes: **Borislav Borissov

---

## [Editor Report · Acceptance letter]

9 Sep 2021

PONE-D-21-09306R1 

Cost-effectiveness analysis of domiciliary topical sevoflurane for painful leg ulcers 

Dear Dr. Gerónimo-Pardo:

I'm pleased to inform you that your manuscript has been deemed suitable for publication in PLOS ONE. Congratulations! Your manuscript is now with our production department. 

Kind regards, 

on behalf of

Dr. Prof. Carlos Alberto Zúniga-González 

Academic Editor

PLOS ONE